# Investigating the Impact of Ultra-Radical Surgery on Survival in Advanced Ovarian Cancer Using Population-Based Data in a Multicentre UK Study

**DOI:** 10.3390/cancers14184362

**Published:** 2022-09-07

**Authors:** Carole Cummins, Satyam Kumar, Joanna Long, Janos Balega, Tim Broadhead, Timothy Duncan, Richard J. Edmondson, Christina Fotopoulou, Rosalind M. Glasspool, Desiree Kolomainen, Simon Leeson, Ranjit Manchanda, Jo Morrison, Raj Naik, John A. Tidy, Nick Wood, Sudha Sundar

**Affiliations:** 1Institute of Applied Health Research, University of Birmingham, Birmingham B15 2TH, UK; 2Coventry and Warwickshire Hospitals NHS Trust, Coventry CV2 2DX, UK; 3Pan Birmingham Gynaecological Cancer Centre, City Hospital, Sandwell and West Birmingham NHS Trust, Birmingham B18 7QH, UK; 4Leeds Teaching Hospitals NHS Trust, Leeds LS1 9LF, UK; 5Norfolk & Norwich University Hospital, Norwich NR4 7UY, UK; 6Division of Cancer Sciences, University of Manchester, Manchester M13 9PL, UK; 7Department of Surgery and Cancer, Faculty of Medicine, Imperial College London, London SW7 2BX, UK; 8Beatson West of Scotland Cancer Centre and University of Glasgow, Glasgow G12 0YN, UK; 9Kings College NHS Foundation Trust, London SE5 9RS, UK; 10Betsi Cadwaladr University Health Board, Bangor LL57 2PW, UK; 11Wolfson Institute of Population Health, Barts Health NHS Trust, London E1 1BB, UK; 12Somerset NHS Foundation Trust, Taunton TA1 5DA, UK; 13Northern Gynaecological Oncology Centre, Gateshead NHS Trust, Gateshead NE9 6SX, UK; 14University of Sheffield, Sheffield S10 2TN, UK; 15Lancashire Teaching Hospitals NHS Foundation Trust, Preston PR7 1PP, UK; 16Institute of Cancer and Genomic Sciences, University of Birmingham, Birmingham B15 2TH, UK

**Keywords:** ovarian cancer, cytoreduction, ultra-radical, radical, surgical complexity, survival, population analysis

## Abstract

**Simple Summary:**

Ovarian cancer is treated by surgery to remove all visible cancer and chemotherapy. Cancer survival is highest when no cancer is left behind after surgery. Ultra-radical (URS) surgery or maximal effort cytoreduction surgery uses additional surgical procedures, e.g., splenectomy, diaphragm stripping, etc. to remove all visible cancer. The use of URS varies internationally with some cancer centres performing this routinely and others not. We conducted a multi-centre study investigating 1471 patients with advanced ovarian cancer (AOC) across three types of gynaecological cancer centres in the UK—those offering mainly low, intermediate or high complexity surgery and investigated survival from cancer in an operated and non-operated whole cohort of women with advanced ovarian cancer. We found that cancer survival was highest in the centres practicing more radical surgery, even after age and deprivation of patients was taken into consideration. Centres practicing mainly low complexity surgery should change practice.

**Abstract:**

We investigated URS and impact on survival in whole patient cohorts with AOC treated within gynaecological cancer centres that participated in the previously presented SOCQER 2 study. National cancer registry datasets were used to identify FIGO Stage 3,4 and unknown stage patients from 11 cancer centres that had previously participated in the SOCQER2 study. Patient outcomes’ association with surgical ethos were evaluated using logistic regression and Cox proportional hazards. Centres were classified into three groups based on their surgical complexity scores (SCS); those practicing mainly low complexity, (5/11 centres with >70% low SCS procedures, 759 patients), mainly intermediate (3/11, 35–50% low SCS, 356 patients), or mainly high complexity surgery (3/11, >35% high SCS, 356 patients). Surgery rates were 43.2% vs. 58.4% vs. 60.9%. across mainly low, intermediate and high SCS centres, respectively, *p* < 0.001. Combined surgery and chemotherapy rates were 39.2% vs. 51.8% vs. 38.3% *p* < 0.000 across mainly low, intermediate and high complexity groups, respectively. Median survival was 23.1 (95% CI 19.0 to 27.2) vs. 22.0 (95% CI 17.6 to 26.3) vs. 17.9 months (95% CI 15.7 to 20.1), *p* = 0.043 in mainly high SCS, intermediate, and low SCS centres, respectively. In an age and deprivation adjusted model, compared to patients in the high SCS centres, patients in the low SCS group had an HR of 1.21 (95% CI 1.03 to 1.40) for death. Mainly high/intermediate SCS centres have significantly higher surgery rates and better survival at a population level. Centres that practice mainly low complexity surgery should change practice. This study provides support for the utilization of URS for patients with advanced OC.

## 1. Introduction

The mainstay of treatment in advanced ovarian cancer is a combination of surgery and platinum-based chemotherapy. The goal of surgery is to achieve complete cytoreduction to nil visible macroscopic residual disease [1,2,3]. The utilization of ultra-radical (URS) or extensive surgery or maximal effort cytoreduction in ovarian cancer offers the potential advantage of achieving complete macroscopic cytoreduction in patients that previously may have been deemed unsuitable for surgery or left with residual disease. URS includes procedures such as diaphragmatic stripping, splenectomy, multiple bowel resections, and resection of porta hepatis nodal disease. As such, these surgical techniques offer the opportunity to overcome the negative aspects of the disseminated disease, also known as poor disease biology. The evidence for this approach derives from case series, comparative studies and post-hoc analysis of chemotherapy trials. Several studies from multiple institutions demonstrate the benefits of this approach with increased surgical clearance and improved survival. These studies tend to compare outcomes to a historical cohort [4,5,6,7]. Some cohort studies compare outcomes between cancer centres [8,9,10]. Post-hoc analysis of chemotherapy trials demonstrates that surgical resection to nil residual disease is associated with better survival [10,11]. The challenge with all these approaches is the selection bias inherent in these types of studies, e.g., patient performance status, willingness to pay and cancer centre volumes are all known confounders for survival. Patients in trials are almost always fitter with better performance status. Population based data analysis reporting outcomes on all patients diagnosed with ovarian cancer offers the advantage of, to a degree, overcoming selection bias associated with reporting surgical centre data by reporting the ‘denominator’ [7,12,13].

It is likely that the gynaecological oncology community no longer has ‘equipoise’ to be able to deliver a randomized controlled trial comparing URS surgery versus lesser complexity surgery for the same disease load. A population-based approach i.e., evaluating outcomes of the whole cohort or denominator of women with Stage3/Stage4/unstaged ovarian cancer can be considered as the next best evaluation. Ovarian cancer care is centralized in the UK with a limited number of gynaecological cancer centres and data from all cancer patients are collected routinely nationally, therefore, population level outcomes can be reported. The principal aim was to assess the role of ultra-radical/extensive surgery in the management of patients with advanced ovarian cancer by evaluating survival and treatment patterns of patients treated in centres with low, intermediate or high surgical ethos. 

## 2. Methods

### 2.1. SOCQER 2 Study

The SOCQER 2 study investigated quality of life (QoL) after low, intermediate and high surgical complexity score surgery in advanced ovarian cancer. The study recruited from 11 cancer centres in England, 1 cancer centre in Wales, 1 cancer centre in Melbourne, Australia and 1 centre in Kolkata, India. Consecutive participants were identified prior to surgical treatment and recruited between September 2015 and September 2016 and followed up to 24 months. Results from the study and description of variation in surgical practice across centres have been previously reported [9]. This paper describes population level outcomes for Stage3/4 and unstaged ovarian cancer patients managed at the 11 participating centres in England. Ethical approval in the UK (West Midlands Solihull UK Research Ethics Committee Reference number 15/WM/0124) included approval for further analysis using routinely collected national data sets for all patients treated within the catchment area of the cancer centres. All cancer centres were of similar case volume, staffed by consultant gynaecological oncologists accredited by the Royal College of Obstetricians and Gynaecologists. In the UK, a national policy dictates that all patients with ovarian cancer within the catchment area should be referred to designated centres for surgical treatment [14].

### 2.2. Defining Centre Surgical Complexity Score Pattern

All patients recruited to the SOCQER2 study had granular data capture for disease load and surgical procedures performed. This was used to derive a surgical complexity score, as per Aletti et al. for each patient. SCS scores of participants in each centre were then used to categorise centres by patterns of surgical practice as centres that practiced mainly low complexity surgery (>70% low SCS use), mainly intermediate surgery (30–50% low SCS use) and mainly high (>35% high SCS use) [15]. The SOCQER 2 study (Sundar et al., BJOG, 2021) recruited a representative sample, approximately 25% of surgically managed Stage 3 and Stage 4 patients across participating centres (range from 10% to 57%) which allowed us to classify the centres into surgical patterns of practice.

Anonymized, routinely collected data from the Cancer Outcomes and Services (COSD), Hospital Episode Statistics (HES) and Systemic Anti-Cancer Therapy Dataset (SACT) were requested from the Office for Data Release for late stage (Stage 3, 4, unstaged) ovarian cancer patients presenting within the referral networks of the English centres participating in the SOCQER 2 study during the recruitment period of SOCQER 2. COSD includes detailed information on tumor topography and morphology, stage, and date of diagnosis. The National Cancer Registration Analysis Service obtains data from multiple sources including hospitals, pathology reports and death certificates and reports in a near 100% case ascertainment [16]. Inclusion of unstaged cases ensured that any variation in the work-up of women with advanced disease and patients with poor prognosis not considered for definitive treatment was captured. An indicator of relative deprivation, Area Income Deprivation, based on the postcode of the patient’s residence, was also included. HES collects data on episodes of inpatient care in English hospitals, including diagnoses coded to ICD-10 and treatment coded to OPCS-4. Data on chemotherapy were contained in the SACT data set. Anonymized patient and tumour identifiers were used to link data for individual patients from these data sets.

Ovary, fallopian tube and primary peritoneal carcinomas were defined as all tumors coded in ICD-10 and ICD-O-2 to C56 (malignant neoplasm of ovary), C57 (malignant neoplasm of other and unspecified female genital organs), C48 (malignant neoplasm of retroperitoneum and peritoneum) excluding certain sarcomas (ICD-O-2 morphologies 8693, 8800, 8801, 8802, 8803, 8804, 8805, 8806, 8963, 8990, 8991, 9040, 9041, 9042, 9043, 9044, 8810, 8811–8921, 9120–9373, 9490, 9500, 9530–9582) and D39.1 (neoplasm of uncertain or unknown behavior of ovary). Only female patients are included. Tumors with borderline or sex stromal or germ cell morphologies were excluded as these patients have a markedly better prognosis than the majority of patients with late stage epithelial ovarian cancers. 

Patients recorded as having stage 3, 4 or unknown stage were included. Stage was that documented at diagnosis, primarily the FIGO stage provided to the cancer registry by the diagnosing trust via multidisciplinary teams. Where missing, the registry utilizes information from pathology reports and clinical investigations to record the most accurate stage at diagnosis possible. Where insufficient data is available, the tumor is considered ‘stage unknown’. 

Treatment of ovarian cancer was defined as the delivery of systemic anti-cancer therapy (‘chemotherapy’) or major surgical resection (‘surgery’) during the primary (i.e., first) course of treatment, defined here as the nine months following diagnosis [17]. Surgical resection was defined on the current COSD definition using the HES OPCS-4 codes.

### 2.3. Statistics

Descriptive statistics with inferential tests appropriate to the distribution were presented. Factors associated with patients being treated with both surgical resection and chemotherapy were explored in a logistic regression analysis. The association between surgical centre treatment policy, other factors and survival were explored in Kaplan–Meier and Cox proportional hazards survival analyses. IBM SPSS Statistics 27 was used for all analyses. 

## 3. Results 

### 3.1. Characteristics of Cohort

A total of 1471 patients across 11 participating centres were identified from September 2015 to September 2016. 

Patients were classified into three groups by the SCS pattern of their centre that had been defined as per the representative sample recruited to the SOCQER 2 study: centres performing >70% low SCS (5/11), mainly low -intermediate SCS surgery (35–50% low SCS) (3/11) and mainly high SCS surgery >35% (3/11) with 788, 365 and 368 patients, respectively (Figure 1 and Figure 2). 

Age and morphology distribution across the cohort were similar with no significant differences between the three SCS groups (Table 1). However, significant differences were noted in deprivation (*p* = 0.002), with the mainly low SCS group having fewer deprived patients. Significant differences in stage were noted (*p* = 0.001), with the mainly low SCS group having more unstaged patients and Stage 3 patients. 

### 3.2. Treatment Patterns

Treatment received (no anticancer treatment/surgery only/chemotherapy only/surgery and chemotherapy) differed significantly across the three Centre SCS groupings (*p* < 0.001) (Table 2). No differences in rates of combined surgery and chemotherapy and considered standard effective treatment were noted between either mainly low (39.9%) or mainly high SCS groups (39%); however, combined treatment rates were higher in the mainly intermediate SCS centres (52%). Rates of patients receiving surgery across the three groups were 43.2% vs. 58.4% vs. 60.9% across low, mainly intermediate and mainly high SCS, respectively (*p* < 0.001). Rates of patients receiving primary debulking surgery followed by chemotherapy increased as surgical radicality increased across the three centre SCS groupings (mainly low 107 (35.3%) vs. mainly intermediate 92 (48.9%) vs. mainly high 78 (56.1%), and, correspondingly, less neoadjuvant chemotherapy followed by interval debulking surgery was performed in mainly low 196 (64.7%) vs. mainly intermediate 96 (51.1%) vs. mainly high 61 (43.9),) (*p* = <0.001). In total, 136 (9.2%) patients received bevacizumab, across mainly low 71 (9.4%), mainly intermediate 39 (11.0%) and mainly high SCS centres 26 (7.3%), *p* = 0.240, ns.

Factors associated with treatment with both surgery and chemotherapy were explored in a logistic regression analysis (Table 3). Women aged 70 to 79 were much less likely (OR 0.301, 95% confidence interval 0.199 to 0.480) and women aged 80 or over were very unlikely to undergo both surgery and chemotherapy (OR 0.052, 95% confidence interval 0.030 to 0.092). Receiving both chemo and surgery (approximation for guideline compliant care) was strongly associated with age (*p* < 0.001).

Patients treated in intermediate SCS centres were more likely to be treated with both surgery resection and chemotherapy (OR 1.83, 95% confidence interval 1.32 to 2.54) compared to patients treated in centres, which carried out mainly low or mainly high SCS surgery. There was no clear association with income deprivation.

### 3.3. Survival

Median survival was 23.1 (95% CI 19.0 to 27.2) vs. 22.0 (95% CI 17.6 to 26.3) vs. 17.9 (95% CI 15.7 to 20.1) months in the mainly high, intermediate and low SCS centres, with a log rank test *p* = 0.043 (Figure 3). In an age and deprivation quintile adjusted by the Cox proportional hazards model, the hazard of death increased steeply with age (4.83, 95% confidence intervals 3.62 to 6.44) in patients aged 80 and over compared with patients aged <50 and in patients whose area of residence was in the highest Area Income Deprivation quintile relative to those in the first deprivation quintile (1.24, 95% confidence intervals 1.04 to 1.40). Compared to patients in the high SCS centres, patients in the low SCS group centres had a hazard ratio of 1.21 (95% confidence intervals 1.03 to 1.40) for death (Table 4). 

## 4. Discussion

This work represents one of the very few population-based multicentre studies evaluating the impact of surgical practice on the survival of patients with advanced epithelial ovarian cancer. The UK is one of the few countries globally that have an established centralized cancer care system, and our data are highly valuable since they report on the entire patient population without many of the selection biases that are well recognized in other non-denominator driven, non-population-based analyses. 

Our study clearly demonstrates that there are significant variations in surgical practice and complexity across the various specialist cancer centres despite the centralized cancer care setting in the UK with centralized guidance and governance. This variation in practice appears to have a significant impact not only on the overall survival, but also on the rate of patients who undergo surgery, the timing of surgery and the rate of patients who manage to achieve a combined systemic and surgical treatment. Through its multicentre population-based design, our analysis reflects real-life practice rather than a per protocol implementation of URS. We discuss the conflicting results from Swedish studies (Falconer, Dahm-Kahler), which may reflect the differences in selection of patients for surgery discuss this later in this section [7,13].

We found that in cohorts balanced for age and morphology, centres with greater utilization of URS increased the proportion of patients treated surgically compared to centres that mainly practiced low complexity surgery. However, centres practicing mainly intermediate SCS were most likely to deliver both surgery and chemotherapy; this may reflect greater use of surgery for palliation, e.g., bowel obstruction in high SCS centres but may also reflect challenges in delivery of chemotherapy after ultra-radical surgery, although this was not observed in the UK-based SOCQER2 study [9].

Centres with a mainly low SCS pattern also show a higher proportion of unstaged and Stage 3 patients. Further research is needed to understand factors contributing to this, which could be differences in approach to women with poor performance status, prehabilitation efforts and population factors including presentation and underlying risk. After adjusting for age and deprivation, women treated in centres that practiced mainly low SCS surgery were more likely to have a poorer survival. 

Centres with mainly low SCS surgery had significantly higher delayed debulking rates compared to centres with higher radicality. We note that four randomized trials have shown equivalence with primary surgery or DDS [18,19] and that results of the TRUST trial in high expertise centres are awaited [20]. Neoadjuvant chemotherapy followed by cytoreductive surgery reduces the extent of surgical procedures necessary to achieve complete cytoreduction; therefore, the difference observed in our study in surgical complexity between mainly intermediate and mainly high complexity centres may be integrally related to the slight difference in primary surgery rates between mainly intermediate and high SCS centres. 

We emphasize that the key finding of our study is that high quality surgery improves survival outcomes even when the whole patient cohort of advanced and unstaged ovarian cancer patients is analysed. Our study was not designed to investigate, nor does it provide insight into the differences in approach between primary surgery or primary chemotherapy and interval surgery. 

Two previous population analyses have been published on Stage 3,4 OC, both from Sweden with conflicting conclusions regarding the utility of Ultra-radical surgery. Falconer et al., [13] published results from a single institution in the Stockholm-Gotland region of Sweden, but are regarded as a population-based analysis, as the Karolinska University Hospital is the only tertiary referral center for gynecologic malignancies in the Stockholm/Gotland Region in Sweden. Here, the authors publish results from a per-protocol implementation of ultra-radical surgery comparing survival outcomes in two cohorts treated in 2008–2011 (364) or 2013–2016 (388). They report a near doubling of complete resection rates paralleled with a reduction in rates of women receiving surgery but no difference in survival in the surgically treated cohort (median survival of 39 months in both cohorts, HR 0.94 (95% CI, 0.75–1.18; *p* = 0.59). Crucially, the protocol did not permit the use of interval surgery after neoadjuvant chemotherapy and only 4% women who were considered unsuitable for primary surgery received interval surgery potentially disadvantaging some women. It is also important to recognize that contrary to the main message of the paper, URS was not introduced in the second cohort. Rather, the proportion of ultra-radical surgery increased between the two study periods, from 15% to 48.8%. Such a study parallels the comparison between the mainly intermediate and the mainly high SCS groups in our study, and the Falconer study may be underpowered to investigate survival differences in these two cohorts. The Dalm-Kahler report [7] on the impact of guideline implementation on 3782 women across Sweden compared cohorts of the same time periods as Falconer. They report an increase in primary surgery rates, an increase in complete cytoreduction and improved relative survival (non-significant) following guideline implementation. 

We believe our study demonstrates the ‘real-life impact’ of increasing radicality. We acknowledge that the optimal proportion of intermediate vs. high SCS surgery for best survival, morbidity and QoL outcomes in advanced ovarian cancer are not known and further population-based studies are needed to establish this. Ideally, research should identify algorithms that can combine clinical and molecular indicators and patient preference in decision aids that can support clinicians, patients and health care systems. Research is also needed in how tumour boards make decisions and how we can safely implement complex surgical interventions in health systems with increasing resource constraints. Models of implementation may include centres with established high complexity centres ‘buddying’ other centres in the learning curve or greater centralisation. 

Ultra-radical surgery has been introduced into clinical care in an effort to achieve tumour clearance in a multiviscerally disseminated disease, where multiple prospective and retrospective studies have shown a clear correlation of survival with residual disease rates [10,21]. Since a study of surgery versus no surgery in the primary presentation of the disease will be ethically almost impossible to achieve, we can learn from the prospective randomised phase III DESKTOP III study, which has demonstrated a clear survival benefit of surgery versus no surgery in the relapsed setting, but only for those patients where complete cytoreduction with no residual disease status was achieved [22]. The surgical information from DESKTOP III has clearly shown that URS techniques are necessary to clear peritoneally disseminated disease, and these patients still benefited significantly from surgery despite their multifocal tumour dissemination patterns. 

Despite this, the actual uptake of URS varies across countries, with one population- based study from the US demonstrating that only 56% of patients with advanced ovarian cancer receive guideline compliant care [23]. Consistent with this, 78% of gynaecological oncologists in the UK and US surveyed reported operating times of 3 h or less on advanced ovarian cancer patients, making it highly unlikely that ultraradical surgery was being routinely delivered [5,24].

Median survival and rates of treatment in our study are lower than in published studies. However, we highlight that comparison with data from individual centres or clinical trials is not appropriate as our data includes unstaged ovarian cancer patients. This group of patients usually present as emergency admissions, and are often too ill for treatment or full diagnostic work-up. Our data also shows that women >80 years are much less likely to be staged and receive treatment. Population-based analyses are scarce in ovarian cancer such that we are simply unable to compare rates across studies. Our methodology is consistent with the UK National ovarian cancer feasibility pilot audit that reports on national treatment rates. 

The strengths of our paper are in the robustness of nationally collected data systems in the UK, where all women diagnosed with ovarian cancer are tracked nationally. Our methods of analysis are aligned with the National Ovarian Cancer Audit [24,25]. We carefully ascertained catchment areas served by the cancer centres to derive total patient cohorts at each cancer centre. Importantly, data are not subject to any modification or selection by individual cancer centres. Thus, selection bias is mitigated. Centralisation of gynaecological cancer care was introduced in the UK in the early 1990s, thus, this study is a true comparison of surgical ethos as participating centres had similar high volume throughput. 

The tax payer-funded health care system, free at the point of delivery in the UK NHS, ensures that almost all patients diagnosed within the catchment area of a cancer centre will receive cancer care within that centre. Movement of patients for second opinions is rare. Although a small percentage of patients will have not been captured who were treated in private care, this is unlikely to change results. 

A further strength is that a significant variation in chemotherapy drugs used across the three SCS patterns is unlikely in the UK NHS because clinical practice is governed by the National Institute and Care Excellence and treatments are available in public health care system free at point of delivery. Therefore, we can more easily filter out the impact of surgical practice on the overall outcome. 

### 4.1. Limitations

We classified centre SCS patterns by analyzing surgical data from patients in each centre recruited to the SOCQER 2 study. Although in each cancer centre, this is just a sample of the cohort, we believe that treatment pattern type is robust because this is consistent with views expressed by the clinicians prior to participation in the study. Although we have evaluated survival by cancer SCS pattern, radicality of surgery may also be a surrogate for other factors such as the expertise of a cancer team to manage clinical risk/salvage patients even when they have poorer performance status; this is certainly suggested by the differential staging patterns seen across the three centre SCS pattern types. We do not know the BRCA status of included patients and this may impact survival differences. The choice of surgery performed is dependent on comorbidity and performance status. Unfortunately, we were unable to adjust for performance status or comorbidity as these data are incomplete in national datasets [17,25]. However we find that deprivation is highest in centres with higher SCS, and as deprivation and comorbidity tend to overlap, it is unlikely that significant differences in morbidity profile exists across the three cohorts that favour the high SCS cohort. On the contrary, it was interesting that even though deprivation was higher in the high SCS/URS cohort, these patients were able to successfully undergo URS that achieved better outcomes. A further limitation is that residual disease data were not available for analysis in this cohort. 

### 4.2. Implications for Practice

The study has implications for UK and international practice. Firstly, UK cancer centres that utilize mainly low SCS procedures should consider changing practice; this will involve investment in staff training and organizational resources such as theatre time, intensive care support and perioperative nursing. Relevance for international practice is that centres can be reassured that careful implementation of URS does not reduce survival compared to centres with very conservative practice. 

## 5. Conclusions

Our population-based results show that, in an age and deprivation adjusted comparison of population-based data, the best survival is achieved at centres with greater radicality/higher complexity surgery compared to centres with high rates of low surgical complexity score surgery. Centres that practice mainly low surgical complexity surgery should consider changing practice. Further research is needed to better understand the selection of women for surgery, including referral thresholds and MDT decision making and the role of patient preference.

## Figures and Tables

**Figure 1 cancers-14-04362-f001:**
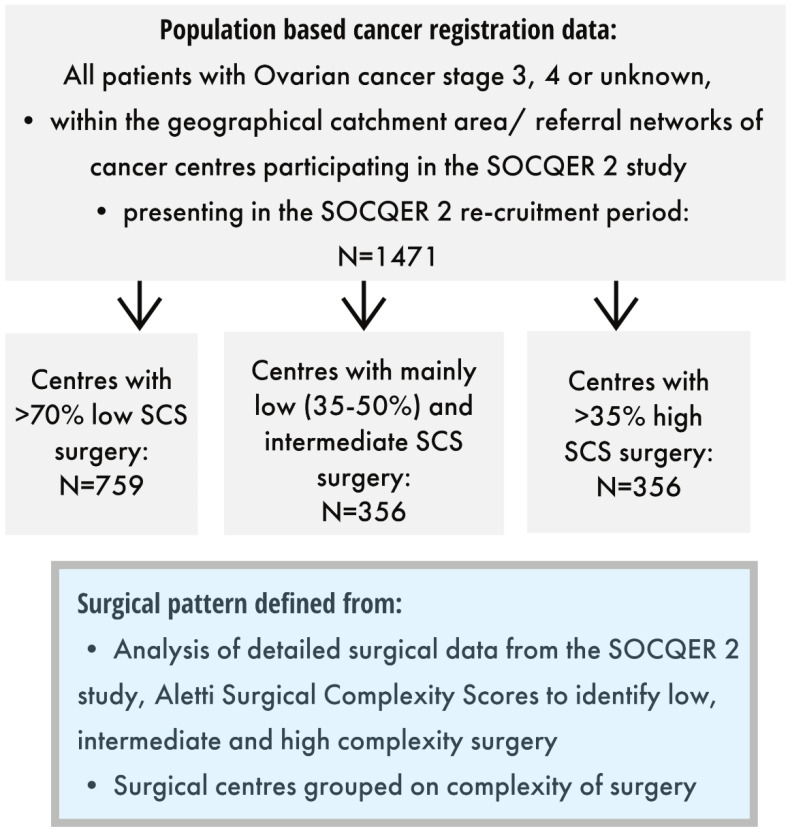
Included patients and surgical centre categories.

**Figure 2 cancers-14-04362-f002:**
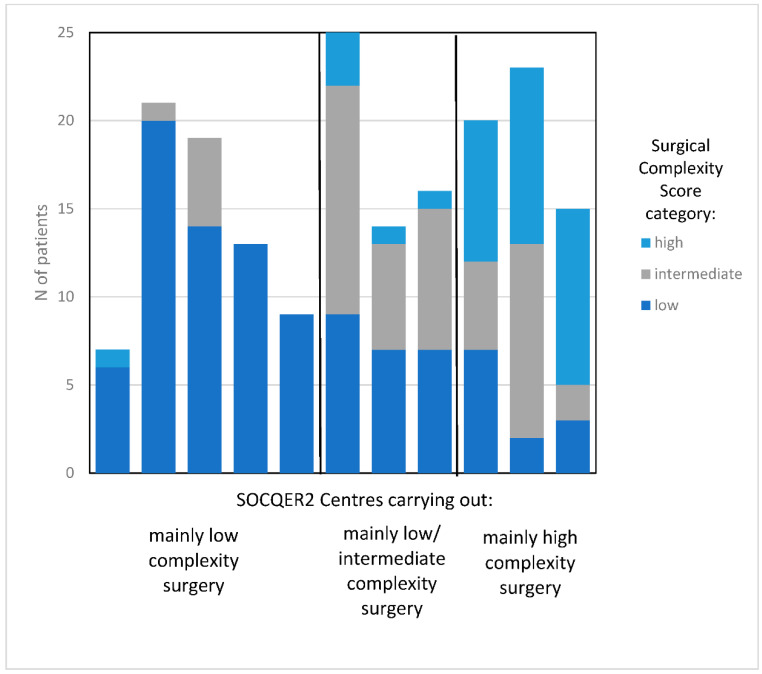
Variation by surgical complexity score in centres participating in the SOCQER2 study.

**Figure 3 cancers-14-04362-f003:**
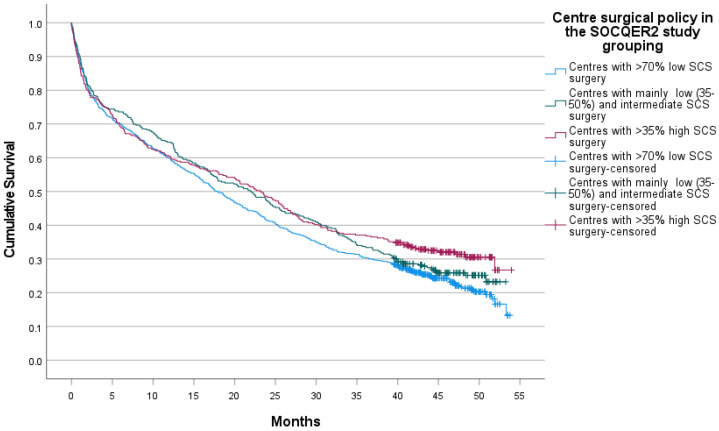
Survival of whole cohort of advanced ovarian cancer patients in centres grouped by surgical policy.

**Table 1 cancers-14-04362-t001:** Patient demographics and disease characteristics and treatment by centre surgical complexity group.

Patient/Disease Characteristics		Centres with >70% Low SCS Surgery		Centres with Mainly Low (35–50%) and Intermediate SCS Surgery		Centres with >35% High SCS Surgery		
		n	%	n	%	n	%	*p*
Age group	0–49	51	6.7%	30	8.4%	29	8.1%	0.369
	50–69	317	41.8%	163	45.8%	140	39.3%	
	70–79	245	32.3%	96	27.0%	122	34.3%	
	80+	146	19.2%	67	18.8%	65	18.3%	
Area Income Deprivation Quintile	1—least deprived	161	21.2%	62	17.4%	54	15.2%	0.002
	2.00	164	21.6%	61	17.1%	56	15.7%	
	3.00	125	16.5%	75	21.1%	92	25.8%	
	4.00	157	20.7%	91	25.6%	85	23.9%	
	5—most deprived	152	20.0%	67	18.8%	69	19.4%	
	Total	759	100.0%	356	100.0%	356	100.0%	
Stage	Unknown	144	19.0%	52	14.6%	49	13.8%	0.001
	3	424	55.9%	182	51.1%	181	50.8%	
	4	191	25.2%	122	34.3%	126	35.4%	
Morphology	Malignant epithelial	667	87.9%	321	90.2%	325	91.3%	0.302
	Miscellaneous and unspecified	87	11.5%	33	9.3%	31	8.7%	
	Non-specific site	5	0.7%	2	0.6%	0	0.0%	

**Table 2 cancers-14-04362-t002:** Analysis of treatment in the whole patient cohort of 1471 women with advanced ovarian cancer by Centre surgical complexity score pattern.

	Centre Surgical Complexity Pattern					
Treatment	Centres with Mainly Low SCS Surgery		Centres with Mainly Intermediate SCS Surgery		Centres with Mainly High SCS Surgery	
	n	%	n	%	n	%
No surgical resection or chemotherapy	217	28.6	94	26.4	93	26.1
Chemotherapy only	214	28.2	54	15.2	83	23.3
Surgical resection only	25	3.3	20	5.6	41	11.5
Surgical resection and chemotherapy	303	39.9	188	52.8	139	39.0
Total	759		356		356	
% in each surgical centre grouping	51.6%		24.2%		24.2%	

**Table 3 cancers-14-04362-t003:** Adjusted Odds Ratios (logistic regression) for undergoing both surgical resection and chemotherapy.

	*p*	Odds Ratio	95% CI	
Age group 0–49 (reference)	<0.001	1		
50–69	0.186	0.749	0.488	1.150
70–79	0.000	0.309	0.199	0.480
80+	<0.001	0.052	0.030	0.092
Area Income Deprivation Quintile 1 least deprived (reference)	0.173	1		
2	0.254	1.236	0.857	1.782
3	0.021	1.167	0.806	1.690
4	0.048	1.057	0.742	1.507
5 most deprived	0.131	0.809	0.562	1.164
Centres with mainly high SCS surgery (reference)	<0.001	1		
Centres with mainly low SCS surgery	0.700	1.057	0.798	1.398
Centres with mainly intermediate SCS surgery	<0.001	1.833	1.322	2.540

**Table 4 cancers-14-04362-t004:** Adjusted hazard ratios for Death.

	HR	95% CI		*p*
Aged <50	1			<0.001
Aged 50–69	1.343	1.019	1.770	0.036
Aged 70–74	2.251	1.704	2.972	<0.001
Aged 80+	4.832	3.624	6.442	<0.001
**Area Income Deprivation Quintile**: 1 (least deprived)	1			0.007
2	0.941	0.773	1.146	0.547
3	0.944	0.777	1.148	0.563
4	1.169	0.970	1.408	0.101
5 (most deprived)	1.241	1.026	1.501	0.026
**SCS score grouping in SOCQER2 study**:				
Centres with mainly high SCS surgery	1			0.054
Centres with mainly intermediate SCS surgery	1.136	0.953	1.354	0.154
Centres with mainly low SCS surgery	1.205	1.036	1.401	0.016

## Data Availability

Restrictions apply to the availability of these data. Data were obtained from the Office for Data Release (now NHS Digital) and to comply with confidentiality and data processing requirements are not available from the authors directly.

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
