# Peer review of "Investigating the Impact of Ultra-Radical Surgery on Survival in Advanced Ovarian Cancer Using Population-Based Data in a Multicentre UK Study"

_cancers, 2022, doi:10.3390/cancers14184362_

Round 1
Reviewer 1 Report
The present study evaluates in a prospectively collected population-base multicenter study the impact of URS in the treatment of patients with advanced OC. The study is of high scientific interest and clinically relevant. The manuscript is concise and the applied methods are adequate.
My comments:
1. please check for some spelling errors throughout the whole manuscript, for example
line 32: "three types"
Minor:
Methods
2. Can you clarify the systemic treatments used in these patients, i.e. rate of bevacizumab use in the whole cohort and in the three subgroups
3. Discussion, line 272
"Despite this, the actual uptake of URS varies across countries, with one population 272 based study from the US demonstrating that only 56% of patients with advanced ovarian cancer receiving guideline compliant care"
please discuss your own data, as less than 55% of patients in all three groups have received surgery and chemotherapy, which would very likely be the guideline compliant care and the impact of IDS and NACT on combination therapy
4. As the results of the present study are in contrast to Falconer et al., and Dahm-Kähler et al., please ellaborate on the differences between these three studies and the reasons for the diverging results
5. The final statement "Centres practicing mainly low complexity surgery should change practice." has to be rephrased in e.g. "The present study supports the concept to provide URS for patients with advanced OC, as URS was associated with improved overall survival".
Author Response
Author responses to all reviewers and the academic editor are in the attached word document.
Reviewer 1
My comments:
- please check for some spelling errors throughout the whole manuscript, for example
line 32: "three types"
Change
This has been corrected
Minor:
Methods
- Can you clarify the systemic treatments used in these patients, i.e. rate of bevacizumab use in the whole cohort and in the three subgroups
136 (9.2%) patients received bevacizumab, low 71 (9.4%), low/intermediate 39 (11.0%), high 26 (7.3%), p=0.240, ns.
Change to manuscript
We have added this results, line 195
- Discussion, line 272
"Despite this, the actual uptake of URS varies across countries, with one population 272 based study from the US demonstrating that only 56% of patients with advanced ovarian cancer receiving guideline compliant care"
please discuss your own data, as less than 55% of patients in all three groups have received surgery and chemotherapy, which would very likely be the guideline compliant care and the impact of IDS and NACT on combination therapy
Response and change to manuscript
We have added the following data to the results, line 213.
Receiving both chemo and surgery (approximation for guideline compliant care) was strongly associated with age (p<0.001). Age is also strongly associated with whether the cancer stage was recorded as unknown. There were no significant differences in the proportion of unstaged according to surgical treatment policy, unstaged stage: mainly low 199 (26.2%), mainly intermediate 80 (22.5%), mainly high 78 (21.9%), p=0.195).
Discussion line 279 reflects the discussion on IDS and NACT.
|
|
- As the results of the present study are in contrast to Falconer et al., and Dahm-Kähler et al., please elaborate on the differences between these three studies and the reasons for the diverging results
Change – We have included the below in Discussion.
Two previous population analyses have been published on Stage 3,4 OC ( Falconer et al and Dalm-Kahler et al), both from Sweden with conflicting conclusions regarding the utility of Ultra-radical surgery. Falconer et al, published results from a single institution in the Stockholm- Gotland region of Sweden, but are regarded as a population based analysis as the Karolinska University Hospital is the only tertiary referral center for gynecologic malignancies in the Stockholm/Gotland Region in Sweden. Here, the authors publish results from a per-protocol implementation of ultra-radical surgery comparing survival outcomes in two cohorts treated in 2008–2011 (364) or 2013–2016 (388). Falconer reports a near doubling of complete resection rates paralleled with a reduction in rates of women receiving surgery but no difference in survival in the surgically treated cohort (median survival of 39 months in both cohorts, HR 0.94 (95% CI, 0.75–1.18; p = 0.59). Crucially, the protocol did not permit the use of interval surgery after neoadjuvant chemotherapy and only 4% women who were considered unsuitable for primary surgery received interval surgery potentially disadvantaging some women. It is also important to recognize that contrary to the main message of the paper, URS was not introduced in the second cohort. Rather the proportion of ultra-radical surgery increased between the two study periods, 15% to 48.8%. As such the study parallels the comparison between the mainly intermediate and the mainly high SCS groups in our study and the Falconer study may be underpowered to investigate survival differences in these two cohorts. Dalm-Kahler report on the impact of guideline implementation on 3782 women across Sweden, comparing cohorts of the same time periods to Falconer. They report an increase in primary surgery rates, an increase in complete cytoreduction and improved relative survival (non significant) following guideline implementation.
We believe our study demonstrates the ‘real-life impact’ of increasing radicality. We acknowledge that the optimal proportion of intermediate vs high complexity surgery or the optimal proportion of women who should receive primary surgery vs interval surgery is currently not known.
- The final statement "Centres practicing mainly low complexity surgery should change practice." has to be rephrased in e.g. "The present study supports the concept to provide URS for patients with advanced OC, as URS was associated with improved overall survival".
Change – we have amended abstract to
Centres that practice mainly low complexity surgery should change practice. This study provides support for the utilization of URS for patients with advanced OC

Reviewer 2 Report
Thank you for submitting the article to the journal.
The article has robust data, appropriate methodology and accurate interpretation.\
Few suggestions:
1. A tubular presentation of a few important similar articles would be helpful for the readers in the discussion section.
2. Any suggestion to researchers, such as the future perspective of the article or any trials on the unanswered questions, should be added.
3 Statistics should be another separate paragraph/ heading.
4. SPSS (company name and location) should be mentioned.
5. Study flowchart is imperative. Please add.
Thank you.
Author Response
Please see document attached
Reviewer 2
Few suggestions:
- A tubular presentation of a few important similar articles would be helpful for the readers in the discussion section.
Added in Discussion, see above.
- Any suggestion to researchers, such as the future perspective of the article or any trials on the unanswered questions, should be added.
Added in Discussion
3 Statistics should be another separate paragraph/ heading. Change This has been added to the manuscript.
- SPSS (company name and location) should be mentioned. Change This has been added to the manuscript.
- Study flowchart is imperative. Please add.
Change .This has been added as Figure 1.
Figure 1. Study illustration shows how cancer centres were categorised in this study

Reviewer 3 Report
It is a very useful study conducted using data from multiple institutions in the UK, enrolling more than 1500 patients and focusing on the Surgical Complexity Score as a surrogate marker of surgical radicality for survival prediction of ovarian cancer patients. It adds relevant data to the already known facts (clear benefits of ultraradical surgery) and contributes to the important discussion whether it is the specialization of the surgeon or the specialization of the center that determines the success of the treatment.
In addition, the study indicates that the surgical complexity score is a suitable surrogate marker for predicting treatment outcomes in ovarian cancer. Paradoxically, the use of the surgical complexity surrogate marker for the quality of surgery can be seen as a limitation of the study. However, the design of this study does not allow for more detailed insights, including the individual expertise of the surgeon and the team. Remarkably, these and all relevant limitations have been correctly addressed and adequately discussed by the authors. As all parts of the manuscript are clear and well structured. The findings of the study are of great clinical importance. I recommend accepting the study for publication.
Author Response
Please see attached response to reviewer 1